# A *Prunus avium* L. Infusion Inhibits Sugar Uptake and Counteracts Oxidative Stress-Induced Stimulation of Glucose Uptake by Intestinal Epithelial (Caco-2) Cells

**DOI:** 10.3390/antiox13010059

**Published:** 2023-12-29

**Authors:** Juliana A. Barreto-Peixoto, Cláudia Silva, Anabela S. G. Costa, Gerardo Álvarez-Rivera, Alejandro Cifuentes, Elena Ibáñez, M. Beatriz P. P. Oliveira, Rita C. Alves, Fátima Martel, Nelson Andrade

**Affiliations:** 1REQUIMTE/LAQV, Department of Chemical Sciences, Faculty of Pharmacy, University of Porto, 4050-313 Porto, Portugal; jpeixoto@ff.up.pt (J.A.B.-P.); claudiasilva@med.up.pt (C.S.); acosta@ff.up.pt (A.S.G.C.); beatoliv@ff.up.pt (M.B.P.P.O.); rcalves@ff.up.pt (R.C.A.); 2Laboratory of Foodomics, Institute of Food Science Research, CIAL, CSIC, Nicolas Cabrera 9, 28049 Madrid, Spain; gerardo.alvarez@csic.es (G.Á.-R.); a.cifuentes@csic.es (A.C.); elena.ibanez@csic.es (E.I.); 3Unit of Biochemistry, Department of Biomedicine, Faculty of Medicine of Porto, University of Porto, 4200-319 Porto, Portugal; 4Instituto de Investigação e Inovação em Saúde (I3S), University of Porto, 4200-135 Porto, Portugal

**Keywords:** cherry stem, infusion, intestinal sugar uptake, antioxidant activity, phenolic compounds

## Abstract

Sweet cherry (*Prunus avium* L.) is among the most valued fruits due to its organoleptic properties and nutritional worth. Cherry stems are rich in bioactive compounds, known for their anti-inflammatory and antioxidant properties. Innumerable studies have indicated that some bioactive compounds can modulate sugar absorption in the small intestine. In this study, the phenolic profile of a cherry stem infusion was investigated, as well as its capacity to modulate intestinal glucose and fructose transport in Caco-2 cells. Long-term (24 h) exposure to cherry stem infusion (25%, *v*/*v*) significantly reduced glucose (^3^H-DG) and fructose (^14^C-FRU) apical uptake, reduced the apical-to-basolateral P_app_ to ^3^H-DG, and decreased mRNA expression levels of the sugar transporters *SGLT1*, *GLUT2* and *GLUT5*. Oxidative stress (induced by *tert*-butyl hydroperoxide) caused an increase in ^3^H-DG uptake, which was abolished by the cherry stem infusion. These findings suggest that cherry stem infusion can reduce the intestinal absorption of both glucose and fructose by decreasing the gene expression of their membrane transporters. Moreover, this infusion also appears to be able to counteract the stimulatory effect of oxidative stress upon glucose intestinal uptake. Therefore, it can be a potentially useful compound for controlling hyperglycemia, especially in the presence of increased intestinal oxidative stress levels.

## 1. Introduction

The influence of plant-derived bioactive compounds on human health has been extensively studied in recent decades, highlighting their impact on human health, as modulators of several pathways involved in the development of several pathological disorders [1,2]. Phenolic compounds are secondary metabolites produced by plants and are known to have anti-inflammatory and antioxidant activity, and can be used to treat and prevent a variety of diseases, such as metabolic and gastrointestinal disorders [3].

Due to its organoleptic properties and nutritional worth, the sweet cherry (*Prunus avium* L.), is among the most valued fruits [4,5]. Cherry leaves, stems and flowers are by-products rich in bioactive compounds; among these, cherry stems, which are a major by-product of cherry harvesting and processing, are still mainly neglected [5]. Despite the fact that these are commonly regarded as a waste product of the food industry, folk medicine recommends them as a traditional herbal remedy, particularly for their diuretic and sedative properties, commonly administered as an infusion or decoction [6]. Even though cherry stem extracts have higher antioxidant activity than cherry extracts, they are still understudied by the scientific community. Nevertheless, they have been shown to have antioxidant and anti-inflammatory properties [4,7]. It is therefore critical to characterize these peduncles’ activities while keeping in mind the bioactive potential of this part of the plant, particularly in the way they are commonly used. In this context, the phenolic profile of four distinct commercial brands of cherry stem infusions was previously characterized by our group, by UHPLC-ESI-QTOF-MS. Forty-four phenolic compounds belonging to eight distinct classes (hydroxybenzoic acids, hydroxycinnamic acids, phenylpropanoic acids, flavan-3-ols, flavonols, flavanones, flavones and isoflavones) were tentatively identified. Interestingly, we also identified salicylic acid for the first time in three of the samples [4]. It is known that these phenolic compounds are the main bioactive compounds responsible for cherry stem infusion’s antioxidant activity, which leads to its antioxidant and anti-inflammatory properties [4,7]. The antioxidant potential of cherry stem infusions from these four distinct commercial brands was also evaluated in this previous study, using 1,1-diphenyl-2-picrylhydrazyl radical (DPPH^•^) scavenging activity, ferric reducing antioxidant power (FRAP) and oxygen radical absorbance capacity (ORAC) tests [4]. Moreover, the antioxidant activity of those infusions was also assessed in a biological system (Caco-2 cells) using the thiobarbituric acid reactive substance (TBARS) assay. Although none of the cherry stem infusions presented antioxidant activity per si, in agreement with its higher chemical antioxidant potential, the commercial brand D cherry stem infusion (but not any of the others) was able to mitigate the increase in malondialdehyde (MDA) levels induced by the oxidative stress inducer TBH in Caco-2 cells [4]. The same work also showed, by using the LDH assay, that the cherry stem infusion D 25% (*v*/*v*) did not exert cytotoxic effects against Caco-2 cells [4], which allows us to safely use it. So, in the present work, the effect of the cherry stem infusion of commercial brand D was further studied. In addition, a phenolic profile characterization of this infusion was performed by UHPLC-ESI-QTOF-MS.

The intestinal tract is commonly exposed to high levels of reactive oxygen species (ROS). Besides being generated at the intestinal level as by-products of normal cellular metabolic activity (namely mitochondrial respiratory chain and the activities of NADPH oxidase, xanthine oxidase, cyclooxygenase, and NO synthase), ROS originate also from exogenous sources such as ultraviolet (UV) radiation, alcohol consumption, cigarette smoking, air pollutants, therapeutical drugs, infections, ischemia-reperfusion injury, some food constituents including sugars, proteins, and lipids, and some food preservatives [8,9,10]. High levels of oxidative stress induce energy metabolism loss, gene alterations and changes in cellular transport systems, cell cycle and cell signaling, and other dysfunctions in intestinal tissue [10,11,12]. Accordingly, various gastrointestinal pathological conditions, including gastrointestinal malignancies, gastroduodenal ulcers, and inflammatory bowel disease (IBD) are associated with increased oxidative stress levels [13]. So, homeostatic regulation of the intestinal oxidative milieu, i.e., maintenance of an equilibrium between the generation of ROS and antioxidant defense mechanisms, is essential to this organ’s function [11,14].

The developing world is seeing a rise in the consumption of high-sugar foods, in particular, high-glucose and fructose foods [15,16,17,18]. Nonetheless, it is commonly recognized that high-sugar diet increases body weight and consequently leads to overweight and obesity, which are a known risk factor for the onset of type 2 diabetes [15,16,17,18]. Glucose is absorbed in the apical membrane of small intestinal epithelial cells via sodium-dependent glucose co-transporter (*SGLT1*) and facilitative glucose transporter 2 (*GLUT2*—a low-affinity glucose transporter) [19,20]. Fructose, on the other hand, is transported across the apical intestinal epithelial cell membrane by using the facilitative glucose transporter 5 (*GLUT5*—a fructose-specific transporter) and *GLUT2* [21,22]. After being taken up by intestinal epithelial cells, fructose and glucose are transported into the portal blood by *GLUT2* [19,20,21,22]. Several studies have shown that some phenolic compounds inhibit glucose and/or fructose intestinal absorption by acting on sugar carriers, which may contribute to their beneficial effect on glucose homeostasis, and hence, help to control hyperglycemia, a characteristic of type 2 diabetes [12,23,24,25,26]. Note that, phenolic compounds are present in high amounts in cherry stems, which are an accessible and low-cost source of such bioactive components [4]. Despite that, there is currently no information on the effect of cherry stem infusion on intestinal sugar uptake. So, this is a good starting point for future studies in this area.

The human epithelial cell line Caco-2 is a useful model widely used to mimic the human intestinal absorptive epithelia in vitro and is thus quite useful for evaluating the bioavailability and potential intestinal effects of bioactives and their metabolites [27]. This model was used in our study to evaluate the impact of a cherry stem infusion on intestinal epithelial glucose and fructose uptake, as well as its ability to regulate the gene expression of glucose (*SGLT1* and *GLUT2*) and fructose (*GLUT5* and *GLUT2*) transporters. Moreover, the involvement of the antioxidant potential of the infusion in these effects was also evaluated.

## 2. Materials and Methods

### 2.1. Samples and Sample Preparation

Cherry stems (*Prunus avium* L.) from a Bulgarian commercial brand was purchased at an herbalist shop in Porto, Portugal. The sample was kept light-protected and at room temperature until the infusion was prepared. The infusions were prepared, in triplicate, according to the manufacturer’s recommendation (1 g of sample in 200 mL of boiled water) in order to mimic the way they are usually prepared and consumed at home. Aliquots of infusions were kept at −21 °C until analysis.

### 2.2. Caco-2 Cell Culture

The Caco-2 (a human colorectal adenocarcinoma cell line) was obtained from ATCC (Manassas, VA, USA) and was used between passage number 40–68. The cells were maintained in a humidified atmosphere of 5% CO_2_–95% air and were grown in a Minimum Essential Medium containing 5.55 mM glucose and supplemented with 15% fetal bovine serum (FBS), 25 mM HEPES and 100 units mL^−1^ penicillin, 100 μg mL^−1^ streptomycin (all from Sigma-Aldrich, St. Louis, MO, USA). The culture medium was renewed every 3–4 days and the culture was split every 12 days. For sub-culturing, the cells were removed enzymatically (0.25% trypsin–EDTA, 5 min, 37 °C), split 1:3, and subcultured in plastic culture dishes (21 cm^2^; ∅ 60 mm; Corning Costar, Corning, NY, USA).

### 2.3. Cell Treatments

To test the effects of cherry stem infusion (250 µL/mL), *tert*-butylhydroperoxide (TBH; 500 µM), N-acetylcysteine (NAC; 1000 µM) or ascorbic acid (500 µM) (all from Sigma-Aldrich), cells were exposed to these compounds for 24 h in FBS-free culture medium or Krebs buffer. Tested compounds were dissolved in H_2_O (cherry stem infusion, NAC and ascorbic acid) or decane (TBH). The final concentration of the solvents in culture medium or buffer was 25% (*v*/*v*) for cherry stem infusion and 1% (*v*/*v*) for TBH, NAC or ascorbic acid.

### 2.4. Determination of ^3^H-Deoxy-D-Glucose and ^14^C-Fructose Apical Uptake

After exposure to cherry stem infusion, TBH, NAC, ascorbic acid, or vehicle for 24 h, cells were washed with 300 µL glucose-free Krebs buffer (containing, in mM: 125 NaCl, 4.8 KCl, 1.2 MgSO_4_, 1.2 CaCl_2_, 25 NaHCO_3_, 1.6 KH_2_PO_4_, 0.4 K_2_HPO_4_, and 20 HEPES (pH 7.4)) at 37 °C. Then, the cell monolayers were preincubated for 20 min in 300 µL Krebs buffer at 37 °C. Uptake was initiated by the addition of 200 µL Krebs buffer at 37 °C containing ^3^H-2-deoxy-D-glucose 10 nM (^3^H-DG; specific activity 60 Ci/mmol) or ^14^C-fructose 100 nM (^14^C-FRU; specific activity 300 Ci/mmol) (both from American Radiolabeled Chemicals, St. Louis, MO, USA). Incubation was stopped after 6 min by removing the incubation medium and rinsing the cells with 500 µL ice-cold buffer. Cells were then solubilized with 300 µL 0.1% (*v*/*v*) Triton X-100 (in 5 mM Tris HCl, pH 7.4) and placed at 4 °C overnight. Intracellular radioactivity was measured by liquid scintillation counting (LKB Wallac 1209 Rackbeta, Turku, Finland). Results were normalized for total protein content (Bradford method).

### 2.5. Evaluation of the Apical-to-Basolateral Transepithelial Transport (Apparent Permeability)

Experiments were carried out using the Caco-2 cells seeded on polyester (PET) microporous filters, in 6-well plates of 1.12 cm^2^ area and 0.4 μm of pore size (Corning Costar). After 15 days of culture, Caco-2 cells were completely differentiated and polarized, resembling the morphological and functional features of the mature enterocytes. Cherry stem infusion or vehicle were properly diluted in Krebs buffer to obtain a final solution concentration of 25% (*v*/*v*). After a 10 min equilibration period, ^14^C-FRU (100 nM) or ^3^H-DG (20 nM) were added to the apical compartment (500 µL) and apical to basolateral transport studies were carried out for 60 min. Every 10 min, 100 μL were withdrawn from the basolateral compartment (1.8 mL) and replaced with an equal volume of the same buffer. Samples (100 μL) were also taken from the donor chamber at the end of each experiment to monitor donor chamber ^14^C-FRU or ^3^H-DG concentration. Samples were analyzed by means of liquid scintillation counting (LKB Wallac 1209 Rackbeta), and results are expressed as apparent permeability (P_app_). At the end of the experiment, the monolayer was fragmented with Triton X-100, for measurement of ^4^C-FRU or ^3^H-DG cellular content.

In order to test the integrity of the monolayer, transport of the paracellular marker phenol red across the surface was determined in each experiment. Phenol red (100 μM) was added to the donor chamber at the beginning of the permeability study, and the amount of phenol red in the acceptor chamber was determined after 60 min. This was performed spectrophotometrically (at 560 nm), using a microplate reader (BioTek instruments, Winooski, VT, USA).

P_app_ was determined according to the following equation:Papp = dQ/dt × 1/(A × Ci) [cm s^−1^]
where dQ/dt (mol s^−1^) is the transport rate, C_i_ (mol cm^−3^) is the initial concentration of ^3^H-DG or ^14^C-FRU in the donor chamber, and A (cm^2^) is the surface area.

### 2.6. Real-Time Quantitative Reverse Transcription PCR

Total RNA was extracted from Caco-2 cells treated with the cherry stem infusion or vehicle for 24 h, and quantitative real-time polymerase chain reaction (qRT-PCR) was carried out as described [28]. The primer pair was 5′-CAG GAC TAT ATT GTG GGC TAA-3′ (forward) and 5′-CTG ATG AAA AGTGCC AAG T-3′ (reverse) for *GLUT2*, 5′-ACC GTG TCC ATG TTT CCA TT-3′ (forward) and 5′-ATT AAG ATC GCA GGC ACG AT-3′ (reverse) for *GLUT5*, and 5′-TGG CAA TCA CTG CCC TTT A-3′ (forward) and 5′-TGC AAG GTG TCC GTG TAA AT-3′ (reverse) for *SGLT1*. The amount of *GLUT2*, *GLUT5*, and *SGLT1* mRNA was normalized to the amount of mRNA of the housekeeping gene, human β-actin. The primer pair for β-actin was: 5′-AGA GCC TCG CCT TTG CCG AT-3′ (forward) and 5′-CCA TCA CGC CCT GGT GCC T-3′ (reverse). Cycling conditions for human *GLUT2*, *GLUT5*, *SGLT1*, and β-actin amplification were the same as those described by Peixoto et al. [28]. Data were analyzed using LightCycler^®^ 96 SW 1.1 analysis software (Roche, Mannheim, Germany), using the Ct method [29]. The β-actin mRNA expression levels were not affected by cell treatment.

### 2.7. Total Protein Determination

The protein content of cell monolayers was determined using human serum albumin as standard, according to the method of Bradford [30].

### 2.8. Analysis of Phenolic Compounds by UHPLC-ESI-QTOF-MS

The phenolic composition of cherry stem infusion was analyzed, as previously described [4], with minor modifications [31]. In brief, after freeze-drying the infusion, the residue was resuspended in methanol/water (50:50, *v*/*v*), filtered, and analyzed in an Agilent 1290 UPLC system (Agilent Technologies, Santa Clara, CA, USA) coupled to an Agilent 6540 quadrupole-time-of-flight mass spectrometer (Q-TOF MS) equipped with an orthogonal ESI source. The compounds were separated at 40 °C using a Zorbax Eclipse Plus C18 column (2.1 × 100 mm, 1.8 μm of particle diameter; Agilent Technologies), with a gradient elution program at a flow rate of 0.5 mL/min and a sample injection volume of 2 μL [4]. The phenolic compounds were identified based on literature data about MS spectra of cherry stems phytochemicals [4,32,33,34], using reference standards to confirm and quantify compounds, when available (gallic acid, chlorogenic acid, caffeic acid, *o*-coumaric acid, *p*-coumaric acid, catechin, quercetin, ferulic acid, luteolin-7-*O*-glucoside, and rutin were all from Sigma-Aldrich (St. Louis, MO, USA), while 4-hydroxybenzoic acid, hesperidin, and genkwanin were from Merck (Darmstadt, Germany)), or to semi-quantify the remaining compounds with structural similarity. The analyses were performed in triplicate and results are expressed as ng of compound/mL of infusion.

### 2.9. Statistical Analysis

Data were expressed as means ± standard error of the mean (S.E.M). n indicates the number of replicates of at least two independent experiments. Statistical significance of the difference between two groups was evaluated by Student’s *t*-test; statistical analysis of the difference between various groups was evaluated by the analysis of variance (two-way ANOVA) test, followed by the Newman-Keuls post-hoc test. Analyses were conducted using the GraphPad Prism version 7.0 software (San Diego, CA, USA). *p* < 0.05 was considered statistically significant.

## 3. Results and Discussion

Due to a “nutrition transition”, the consumption of high-sugar foods is increasing throughout the developing world and has been unquestionably related to the global obesity and diabetes epidemics [15,18]. High-glucose and fructose diets may be an important contributor to this pandemic, as excessive consumption of both sugars appears to lead to insulin resistance and obesity, which, consequently, can result in the development of type 2 diabetes [15,16,17,18]. However, the biological mechanisms underpinning this link remain unknown. Fructose and glucose have different metabolic fates and absorption rates, so they are expected to have different effects on the development of specific features of obesity and/or type 2 diabetes. Glucose transport is an energy-requiring process mediated by *SLGT1* and *GLUT2* [19,20], whereas fructose moves through a facilitated passive transport mainly mediated by *GLUT5* [21,22]. Research employing several phytochemicals or food matrices rich in these bioactives supports an inhibitory effect on sugar uptake in an intestinal context [12,23,24,25,26]. Some of these phytochemicals are found in cherry stem extracts [4] and have demonstrated biological activities such as antioxidant [4,34,35] and antidiabetic proprieties [34,35], as will be further discussed in Section 3.4.

### 3.1. Cherry Stem Infusion Reduces ^3^H-DG and ^14^C-FRU Apical Uptake by Caco-2 Cells

The effect of the cherry stem infusion on cellular glucose and fructose uptake by Caco-2 cells was investigated by using ^3^H-DG and ^14^C-FRU, respectively. Our results show that cherry stem infusion (25% (*v*/*v*)) significantly inhibits cellular ^3^H-DG and ^14^C-FRU uptake, being the inhibition more pronounced in relation to ^3^H-DG uptake (~45%) than in relation to ^14^C-FRU uptake (~20%) (Figure 1a,b). Of note, this effect is not related to a cytotoxic effect, as we have previously confirmed that cherry stem infusion (25% (*v*/*v*)) does not affect cell viability [4]. These results suggest this infusion may have antidiabetic activity, given its efficacy in inhibiting both glucose and fructose intestinal epithelial uptake.

The inhibitory effect of the cherry stem extract is most probably related to its polyphenolic content, as several dietary polyphenols and polyphenol-rich foods and extracts are known to interfere with the intestinal transport of glucose and/or fructose by a specific effect on sugar transporters [36,37].

### 3.2. Cherry Stem Infusion Reduces the Apical-to-Basolateral P_app_ to ^3^H-DG in Caco-2 Cells

Next, we determined whether cherry stem infusion interferes with the apical-to-basolateral intestinal epithelial permeability to glucose and fructose by evaluating its effect on the P_app_ of Caco-2 cells grown in Transwells to ^3^H-DG and ^14^C-FRU. We verified that the addition of cherry stem infusion to the apical compartment decreased by about 25% the apical-to-basolateral P_app_ to ^3^H-DG (Figure 2a). On the other hand, incubation of Caco-2 cells with cherry stem infusion did not result in a significant alteration in the apical-to-basolateral P_app_ to ^14^C-FRU (Figure 2b). Notwithstanding, and confirming our previous experiments, the cherry stem infusion significantly reduced the amount of ^3^H-DG and ^14^C-FRU in the Caco-2 cell monolayer (Figure 2c,d). Notably, these results suggest that the cherry stem infusion diminishes the intestinal permeability to glucose but not to fructose in Caco-2 cells. These results further confirm the potential antidiabetic activity of the extract.

### 3.3. Cherry Stem Infusion Affects the Expression of Glucose and Fructose Intestinal Transporters

In the intestine, an uptake of glucose is initiated at the apical membrane by the cotransporter *SGLT1* and by the facilitative *GLUT2*, being glucose then exported across the basolateral membrane through *GLUT2* [19,20]. Regarding fructose, its apical uptake involves the facilitative *GLUT5* (and also *GLUT2*) and this sugar is exported into the systemic circulation by *GLUT2*, located at the basolateral membrane [21,22]. Hence, levels of *SGLT1*, *GLUT2*, *GLUT5* are critical for the process of glucose and fructose intestinal absorption [19,20,21,22].

So, we decided to evaluate the impact of the cherry stem infusion on the gene expression levels of the main glucose (*SGLT1* and *GLUT2*) and fructose (*GLUT5* and *GLUT2*) transporters, that could explain the reduction in ^3^H-DG and ^14^C-FRU uptake and in the P_app_ to ^3^H-DG. Interestingly, a very marked decrease in *SGLT1* (~76%), *GLUT2* (~63%) and *GLUT5* (~48%) mRNA levels was observed in cherry stem infusion-treated cells (Figure 3). These findings are consistent with recent studies describing that several dietary polyphenols reduced intestinal *SGLT1* [25,38,39], *GLUT2* [24,25,26,38] and *GLUT5* [22,26] mRNA levels. Of importance, *GLUT2* was detected not only on the basolateral side, but also on the apical side of membrane of Caco-2 cells [40]. So, similarly to *SGLT1* and *GLUT5*, it can be directly exposed to the infusion.

Taken together, these findings allow us to conclude that cherry stem infusion downregulates the expression of the three main sugar transporters, resulting in a reduction in glucose and fructose intestinal epithelial uptake. Moreover, these results suggest that cherry stem infusion, by decreasing the intestinal absorption of glucose and fructose, has a potential beneficial effect in the context of obesity and type 2 diabetes.

### 3.4. The Inhibitory Effect of the Cherry Stem Infusion upon the Uptake of ^3^H-DG Is Related to Its Antioxidant Activity

In the final set of experiments, we investigated if the inhibitory effect of the cherry stem infusion on ^3^H-DG uptake (which was the most potent inhibitory effect of the infusion) by Caco-2 cells is associated with its antioxidant effect.

For this, Caco-2 cells were treated with a pro-oxidant inducer (TBH) and the effect of the cherry stem infusion on the alterations of ^3^H-DG uptake induced by TBH were evaluated. In these experiments, two known antioxidant agents (NAC and ascorbic acid) were also used as a positive antioxidant control. TBH caused an increase in ^3^H-DG uptake, in agreement with our previous work [14]. In our experimental conditions, the reduction of ^3^H-DG uptake by Caco-2 cells promoted by cherry stem infusion was comparable to that demonstrated by NAC and ascorbic acid. Moreover, cherry stem infusion was able to abolish the effect of TBH on ^3^H-DG uptake to an extent comparable to that of ascorbic acid and NAC (Figure 4). Interestingly, we recently verified that the cherry stem infusion does not present antioxidant activity *per si*, but is able to mitigate the increase in malondialdehyde (MDA) levels induced by TBH in these cells [4]. So, the cherry stem infusion appears to protect Caco-2 cells from oxidative stress-induced increase in glucose uptake.

### 3.5. Quantification of Phenolic Compounds in the Cherry Stem Infusion by UHPLC-ESI-QTOF-MS

As mentioned, our previous work showed that the cherry stem infusion displayed protective properties against oxidative stress-induced lipid peroxidation in Caco-2 cells [4]. Lipid peroxidation can induce changes in biological membranes functions such as permeability, structure, or even fluidity [41]. It is widely accepted that polyphenolics such as phenolic acids or flavonoids are responsible for the majority of the biological properties present in cherries and their by-products. Indeed, a wide variety of cherry stem extracts have demonstrated (by our [4] and by other groups [32,33,34,42,43,44]) a significant antioxidant activity, associated with their contents in phenolic acids and flavonoids. According to literature data, 3-*O*-caffeoylquinic acid, *p*-coumaroylquinic acid, caffeoylquinic acid glucoside, ferulic acid glucoside, *p*-coumaric acid, caffeic acid, and gallic acid are the phenolic acids most commonly found in cherry stems [4,33,34,45], while dihydrowogonin-*O*-glucoside, sakuranetin-*O*-glucoside, chrysin-7-*O*-glucoside, aromadendrin-7-*O*-hexoside, kaempferol 3-*O*-rutinoside-*O*-hexoside, quercetin 3-*O*-rutinoside, naringenin 7-*O*-glucoside, taxifolin-*O*-glucoside, genistein-7-*O*-glucoside, and catechin are the flavonoids most commonly detected [4,33,34,45].

The phenolic profile of the cherry stem infusion herein studied is described in Table 1. Chlorogenic acid, caffeic acid, *p*-coumaric acid, ferulic acid, gallic acid, rutin, and catechin were identified and quantified with authentic standards, while for the remaining compounds, a semi-quantification was made using reference standards with a structural similarity of the tentatively annotated compounds. Our results are in accordance with literature data: the hydroxycinnamic acids *cis*-3-*O*-*p*-coumaroylquinic acid, 3-*O*-caffeoylquinic acid (chlorogenic acid), and *trans*-3-*O*-*p*-coumaroylquinic acid together with the flavanones dihydrowogonin-*O*-glucoside/sakuranetin-*O*-glucoside and dihydrowogonin/sakuranetin were the major phenolics, although other compounds such as chlorogenic acid glucoside, gallic acid, naringenin-7-*O*-glucoside (prunin), naringenin-*O*-glucoside, aromadendrin-*O*-glucoside, quercetin-*O*-rutinoside-*O*-hexoside, kaempferol-*O*-rutinoside-*O*-hexoside, taxifolin-*O*-glucoside, quercetin-*O*-rutinoside, kaempferol-*O*-rutinoside, and chrysin-*O*-glucoside were also detected in high amounts. Overall, cherry stem infusion can be considered a good source of phenolic compounds, which have been described as the main responsible for the bioactive properties of cherry stems.

### 3.6. The Antidiabetic Potential of Cherry Stem Infusion

Evidence from clinical trials showed that phenolic-rich diets stimulate glucose metabolism in individuals at high risk of type 2 diabetes [46]. When cherries are specifically analyzed, a similar trend was found, as the ingestion of cherry juice significantly reduced fasting blood glucose and levels of glycated hemoglobin (HbA1c) in diabetic humans [47]. In in vitro and animal studies, antidiabetic effects have also been observed. Indeed, phenolic fractions from sweet cherry (anthocyanin, hydroxycinnamic acid, and flavonol-rich fractions) were able to promote cellular glucose consumption by HepG2 cells [48]. Similarly, quercetin and kaempferol (two flavonoids found in our infusion in derivative forms) improved glucose uptake in HepG2 [49] and 3T3-L1 [50] cells, suggesting that these phenolic compounds increase glucose conversion to glycogen in the liver and enhance glucose storage by adipocyte cells. In addition, in diabetic mice, a diet supplemented with phenolic compounds present in cherries reduced fasting glycaemia and attenuated the increase of plasma glucose [51,52]. Also, cherries and cherry stems and leaves and flowers were found to inhibit α-glucosidase activity [34,35]. Interestingly, cherry stems possessed a more potent anti-α-glucosidase activity than leaves or flowers [34].

In the present work, we verified that the cherry stem infusion reduces ^3^H-DG and ^14^C-FRU apical uptake and the apical-to-basolateral P_app_ to ^3^H-DG in Caco-2 cells. Moreover, our findings suggest that the inhibitory effect of cherry stem infusion on glucose and fructose uptake appears to be mediated by a negative modulation of glucose (*SGLT1* and *GLUT2*) and fructose (*GLUT5* and *GLUT2*) transporter expression. The stronger inhibitory effect of the cherry stem on glucose transporters (*SGLT1* (~76%) and *GLUT2* (~63%)), when compared with the fructose transporters (*GLUT5* (~48%), and *GLUT2* (~63%)) may well be the reason why only the apical-to-basolateral P_app_ to ^3^H-DG was significantly reduced by the infusion.

Although the effect of cherry stem on glucose and fructose intestinal absorption was not previously investigated, some phenolic compounds found in our infusion in free and/or glycosylated forms (namely quercetin, quercetin glucoside, catechin, epicatechin, caffeic acid, ferulic acid, coumaric acid, and chrysin) were previously found to reduce intestinal glucose [12,23,24,25,39] and fructose [22,26,53] absorption (reviewed in [36,37]). Moreover, our observation that the cherry stem infusion decreases *SGLT1*, *GLUT2* and *GLUT5* mRNA expression levels is in line with previous research which report that polyphenols are able to decrease the expression of these intestinal glucose and fructose transporters [36]. Moreover, some interactions between flavonoids (in particular quercetin glucoside) and *SGLT1* [54] and *GLUT2* [24] have been reported, suggesting that they might compete with glucose for these transporters. Also, caffeoylquinic acid (a major compound present in our infusion) was identified as able to inhibit *SGLT1* and *GLUT2* in intestinal cells [25], decreasing the absorption of glucose by Caco-2 cells. Similarly, polyphenols such as catechin [55], caffeic acid [26], kaempferol [56] and quercetin [26,53] were demonstrated to inhibit *GLUT5* and *GLUT2*, delaying the absorption of fructose. Since these substances are present in cherry stem infusion, blockade of these glucose and fructose transporters by these phenolic compounds will reduce the absorption of these sugars by the intestinal epithelial cells, helping to reduce the postprandial increase in blood glucose levels.

In the final part of the work, we were able to establish that the inhibitory effect of the cherry stem infusion upon ^3^H-DG uptake is related to its antioxidant activity. This is very interesting, as it shows a direct link between the cellular oxidative status and intestinal transporters expression and activity, an area where knowledge is still very limited [57,58]. In this context, it should be noted that ROS, generated locally by tissue, regulate post-transcriptional changes in several mRNA families. This regulation is achieved by the turnover and translation of regulatory mRNA-binding (TTR-RBP) proteins, which regulate the degree of gene expression in response to oxidative damage. So, lower rates of sugar uptake following cherry stem infusion exposure may be partially explained by the possibility that transporters such as *SGLT1*, *GLUT2,* and *GLUT5* are among the target genes controlled by TTR-RBP and that their post-transcriptional processing is susceptible to oxidative stress [59].

Because type 2 diabetes is associated with increased oxidative stress levels [60], the fact that the cherry stem infusion was able to abolish the stimulatory effect of oxidative stress upon ^3^H-DG uptake suggests that this infusion may be particularly interesting for decreasing intestinal glucose absorption in diabetic individuals.

There is currently limited information in the literature about the mechanism underlying the inhibition of glucose and fructose transport across Caco-2 cells by cherries and their by-products. As it stands, our work appears to be the first to study the potential consequences of this relationship; however, our findings validate the biological significance of polyphenols and suggest that when administered in combination, as with cherry stem infusion, they might be more biologically active.

## 4. Conclusions

Overall, the data presented in this study indicate that cherry stem infusion is capable of efficiently reducing glucose and fructose apical uptake by Caco-2 intestinal cells and glucose transport across these cells by downregulating the most important glucose transporters (*SGLT1* and *GLUT2*), and the principal fructose transporter (*GLUT5*). In addition, this natural product mitigates the effect of oxidative stress upon glucose uptake. Importantly, all these interesting outcomes can be linked to the antioxidant properties displayed by cherry stem infusion.

Inhibition of glucose and fructose uptake in the small intestine contributes to the regulation of blood glucose levels, particularly post-prandial hyperglycemia, a characteristic of obesity and type 2 diabetes. It is important to mention that increased glucose absorption can be a result of increased oxidative stress at the intestinal level, and in this context, the cherry stem infusion can mitigate the effects of oxidative stress, thus consequently lowering the risk of obesity and diabetes. In this way, we can speculate that regular dietary consumption of cherry stem infusion may exert a protective antioxidant effect at the intestinal epithelial level, resulting in a reduction in sugar absorption, thus enhancing glycemic management by restricting or delaying intestinal glucose and fructose absorption. Therefore, it can be a potentially useful alternative/beverage for controlling hyperglycemia, especially when associated with increased intestinal oxidative stress levels

## Figures and Tables

**Figure 1 antioxidants-13-00059-f001:**
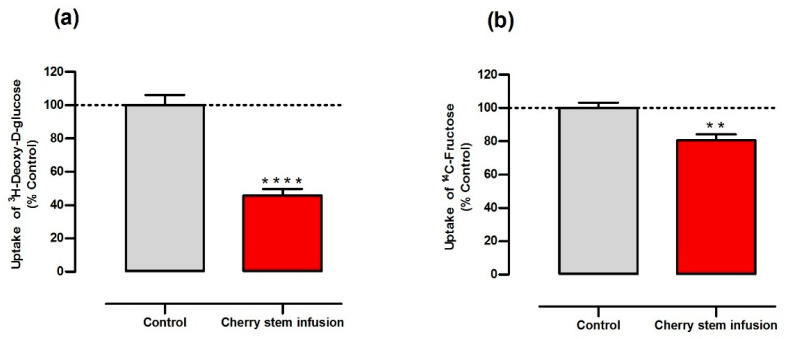
Effect of cherry stem infusion on the uptake of (**a**) ^3^H-DG and (**b**) ^14^C-FRU by Caco-2 cells. Caco-2 cells were exposed for 24 h to the cherry stem infusion (250 µL/mL) or the respective vehicle (control) (n = 10). Data are shown as mean ± S.E.M., by Student’s *t*-test. ** *p* < 0.01; **** *p* < 0.0001 significantly different from control.

**Figure 2 antioxidants-13-00059-f002:**
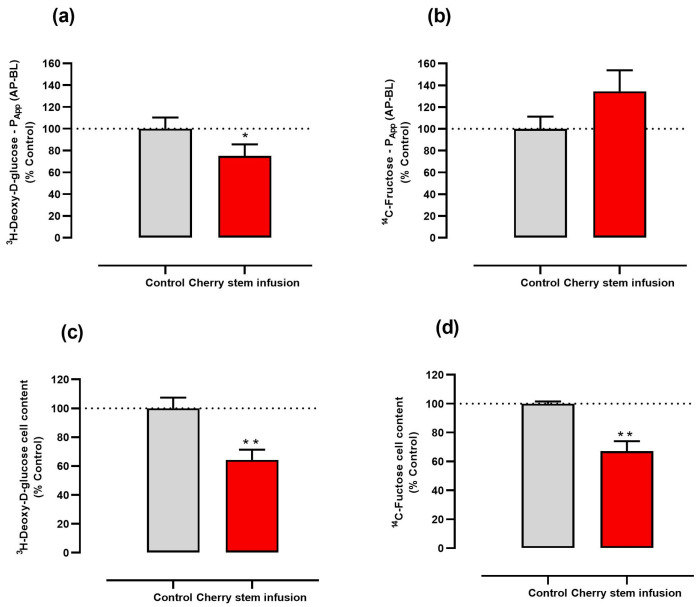
Effect of cherry stem infusion on the apical-to-basolateral (AP−BL) apparent permeability (P_app_) to (**a**) ^3^H-DG and (**b**) ^14^C-FRU and on the cellular content of (**c**) ^3^H-DG and (**d**) ^14^C-FRU (n = 6–7) Data are shown as mean ± S.E.M., by Student’s *t*-test. * *p* < 0.05; ** *p* < 0.01 significantly different from control.

**Figure 3 antioxidants-13-00059-f003:**
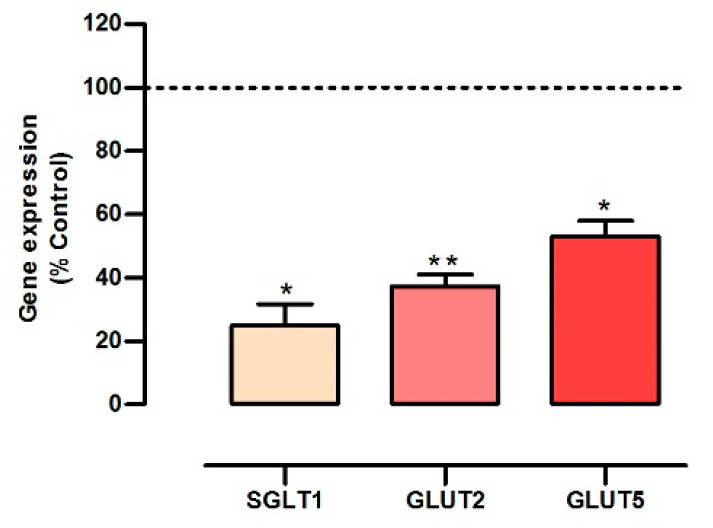
Effect of cherry stem infusion on sodium-dependent glucose cotransporter (*SGLT1*), facilitative glucose transporter 2 (*GLUT2*) and facilitative glucose transporter 5 (*GLUT5*) mRNA levels in Caco-2 cells (n = 6–7). Data were normalized to the expression of β-actin. Data are shown as mean ± S.E.M., by Student’s *t*-test. * *p* < 0.05; ** *p* < 0.01 significantly different from control.

**Figure 4 antioxidants-13-00059-f004:**
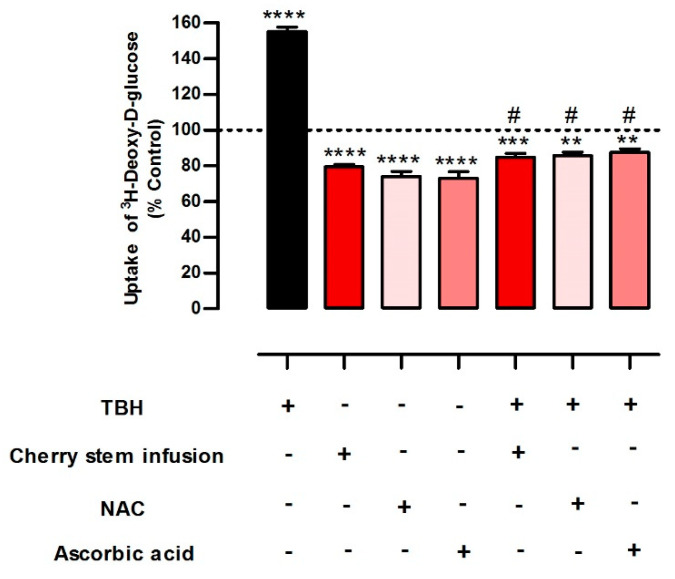
Effect of cherry stem infusion, N−acetylcysteine (NAC) and ascorbic acid, on the uptake of ^3^H-DG by Caco-2 cells, in the presence and absence of *tert*−butylhydroperoxide (TBH) (n = 8). Uptake was measured by incubating Caco-2 cells at 37 °C with ^3^H-DG (10 nM) for 6 min. Data are shown as mean ± S.E.M., by two-way ANOVA with Newman-Keuls post hoc test. ** *p* < 0.01; *** *p* < 0.001; **** *p* < 0.0001 significantly different from control; # *p* < 0.01 significantly different from TBH.

**Table 1 antioxidants-13-00059-t001:** Quantification of compounds detected in cherry stem infusion by UHPLC-ESI-QTOF-MS (ng of compound/mL of infusion).

Family	Compound	Std *	Concentration (ng/mL of Infusion)
*Hydroxycinnamic acids*	Chlorogenic acid glucoside	A	928.9 ± 59.0
	*cis*-Caffeic acid 4-glucoside (*cis*-Glucocaffeic acid)	B	56.5 ± 5.3
	*cis*-3-*O*-*p*-Coumaroylquinic acid	A	15,590.5 ± 1718.7
	*trans*-Caffeic acid 4-glucoside (*trans*-Glucocaffeic acid)	B	58.8 ± 3.9
	3-*O*-Caffeoylquinic acid (Chlorogenic acid)	A	9421.4 ± 1080.0
	*beta*-D-Glucosyl-2-coumarate (*cis*-Melilotoside)	C	511.6 ± 243.2
	Caffeic acid	B	292.5 ± 44.7
	1-*O*-*p*-Coumaroyl-beta-D-glucose	D	60.4 ± 7.8
	*trans*-*p*-Coumaric acid glucoside	D	99.1 ± 20.0
	*trans*-3-*O*-*p*-Coumaroylquinic acid	A	9195.7 ± 398.1
	Ferulic acid glucoside	E	323.5 ± 32.1
	*trans*-*o*-Coumaric acid glucoside	C	454.3 ± 30.8
	*p*-Coumaric acid	D	103.9 ± 7.4
	Ferulic acid	E	99.3 ± 3.4
*Hydroxybenzoic acids*	Gallic acid	F	1704.5 ± 150.0
	Hydroxybenzoic acid glucoside	G	21.0 ± 1.5
	Salicylic acid	G	91.7 ± 0.9
*Phenylpropanoic acids*	3-(2-Hydroxyphenyl)propionic acid glucoside	C	1314.2 ± 112.8
	3-(2-Hydroxyphenyl)propionic acid (Melilotic acid)	C	556.3 ± 71.8
*Flavanones*	Dihydroxymethoxy flavanone-O-pentosylhexoside (I)	H	144.1 ± 16.9
	Tetrahydroxyflavanone-glucoside	H	406.8 ± 53.5
	Naringenin-7-*O*-glucoside (Prunin)	H	1515.9 ± 129.0
	Dihydroxyflavanone-*O*-pentosylhexoside	H	112.1 ± 12.1
	Dihydroxymethoxy flavanone-*O*-pentosylhexoside (II)	H	90.0 ± 8.5
	Naringenin-*O*-glucoside	H	1599.9 ± 136.0
	Dihydrowogonin/Sakuranetin (I)	H	2326.5 ± 60.8
	Dihydrowogonin-*O*-glucoside/Sakuranetin-*O*-glucoside (I)	H	6620.2 ± 384.2
	Dihydrowogonin/Sakuranetin (II)	H	5651.5 ± 328.6
*Flavonols*	Aromadendrin-*O*-glucoside	I	1184.5 ± 108.6
	Quercetin-*O*-rutinoside-*O*-hexoside	I	1789.1 ± 219.7
	Kaempferol-*O*-rutinoside-*O*-hexoside	I	2006.8 ± 19.8
	Taxifolin-*O*-glucoside	I	1355.3 ± 124.1
	Quercetin-3-*O*-rutinoside (Rutin)	J	2200.3 ± 257.0
	Quercetin-3-*O*-glucoside	I	560.2 ± 12.0
	Kaempferol-3-*O*-rutinoside (Nicotiflorin)	I	3869.3 ± 446.9
	Kaempferol-3-*O*-glucoside (Astragalin)	I	773.6 ± 35.1
*Flavones*	Tetrahydroxyflavone hexoside (I)	K	363.1 ± 171.3
	Tetrahydroxyflavone hexoside (II)	K	740.6 ± 349.1
	Chrysin-7-*O*-glucoside isomer	K	296.0 ± 142.0
	Chrysin-7-*O*-glucoside	K	2357.1 ± 1116.5
*Flavan-3-ols*	Catechin	L	84.5 ± 12.5
	Epicatechin	L	24.3 ± 1.7
*Isoflavones*	Genistein-*O*-glucoside	K	341.1 ± 160.8
	Methyl genistein (Prunetin)	M	2.4 ± 1.1

Results are expressed as mean ± standard deviation (ng/mL of infusion) of the infusion prepared in triplicate. *, Reference standard used for quantification or semi-quantification: A: Chlorogenic acid; B: Caffeic acid; C: *o*-Coumaric acid; D: *p*-Coumaric acid; E: Ferulic acid; F: Gallic acid; G: 4-hydroxybenzoic acid; H: Hesperidin; I: Quercetin; J: Rutin; K: Luteolin-7-glucoside; L: Catechin; M: Genkwanin.

## Data Availability

The data presented in this study are available on request from the corresponding author.

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
