# Peer review of "A Prunus avium L. Infusion Inhibits Sugar Uptake and Counteracts Oxidative Stress-Induced Stimulation of Glucose Uptake by Intestinal Epithelial (Caco-2) Cells"

_antioxidants, 2023, doi:10.3390/antiox13010059_

Round 1
Reviewer 1 Report
Comments and Suggestions for Authors
Manuscript ID: antioxidants-2771133
“A Prunus avium L. infusion inhibits sugar uptake and counteracts oxidative stress-induced stimulation of glucose uptake by intestinal epithelial (Caco-2) cells”
The manuscript includes very interesting content. The study attempted to investigate at the cellular level the impact of cherry stem infusion on intestinal epithelial glucose and fructose uptake, as well as its ability to regulate the gene expression of glucose (SGLT1 and GLUT2) and fructose (GLUT5 and GLUT2) transporters. Moreover, the involvement of the antioxidant potential of the infusion in these effects was also evaluated.
Line 139- 140: What are the concentrations of TBH, NAC and ascorbic acid?
Line 145: “Determination of 3H-deoxy-D-glucose and 14C-fructose and 3H-deoxy-D-glucose” – correct it
Line 151: "Why were different concentrations of these compounds applied?"
Line 166: “3H-DG (20 nM)” - Why is the HDG concentration different here than in line 151?
Line 183: “…the initial concentration…” – the initial concentration of what?
Line 259-261; 302-304; 342-343; 375-377: "I think that this is included in the methodology and unnecessary here."
3.2. "There is a lack of discussion."
Line 376: Add concentration of NAC, ascorbic acid and TBH?
Author Response
Thank you very much for your evaluation of our manuscript.
Please see next our answers to your comments.
Best regards.
Fátima Martel, corresponding author
Line 139- 140: What are the concentrations of TBH, NAC and ascorbic acid?
Answer: There was a mistake in the text (µl instead of µM), now the concentrations are shown.
Line 145: “Determination of 3H-deoxy-D-glucose and 14C-fructose and 3H-deoxy-D-glucose” – correct it
Answer: corrected. Thank you.
Line 151: "Why were different concentrations of these compounds applied?"
Answer: 3H-DG and 14C-FRU were applied in different concentrations because if we used the same amount of 3H-DG and 14C-FRU (10 nM), the cpm values (scintillation countings per min) for 14C-FRU would be too low, and it would not be possible to correctly quantify 14C-FRU uptake. We used 10 nM 14C-FRU in the first series of experiments, then we verified this, so we decided to use a higher concentration (100 nM).
Line 166: “3H-DG (20 nM)” - Why is the HDG concentration different here than in line 151?
Answer: Because of the same reason stated in the previous comment. If we have used 10 nM 3H-DG in the transwell experiments, the cpm values (scintillation countings per min) would be rather low, and we could not correctly quantify 3H-DG Papp. Please note that in th Transwell experiments, we measured transcellular transport. For this, we placed 3H-DG in the apical chamber and measured 3H-DG in the basolateral chamber. That is why the cpm values are low.
Line 183: “…the initial concentration…” – the initial concentration of what?
Answer: this information was added.
Line 259-261; 302-304; 342-343; 375-377: "I think that this is included in the methodology and unnecessary here."
Answer: Deleted as suggested, in all the Figure legends.
3.2. "There is a lack of discussion."
Answer: In agreement with your comment, discussion concerning the results presented in section 3.2 (and their relation to the other results) was added to the manuscript (lines 417-424).
Line 376: Add concentration of NAC, ascorbic acid and TBH?
Answer: Following your criticism, this part of the legend was removed, but the concentrations are shown in the Methods section.
Reviewer 2 Report
Comments and Suggestions for Authors
I have carefully reviewed your manuscript and would like to offer constructive feedback for improvement. Overall, the study conducted by Barreto-Peixoto et al. on the potential effect of cherry stems (from Prunus avium L.) on sugar uptake in the Caco-2 cell line is intriguing. However, there are some concerns that need to be addressed to strengthen the manuscript:
1. The use of q-PCR to assess glucose transporter mRNA expression is noted. However, to comprehensively explain the glucose uptake phenotype, it is recommended to include analysis at the protein level or evaluate transporter translocation. This additional information would enhance the study's depth and clarity.
2. Fig 4 lacks supportive data, presenting a significant issue in the manuscript. It is crucial to address this concern by including relevant information. Specifically, the absence of ROS level data after different treatments should be rectified. Without this data, it is challenging to draw conclusions about the observed lower sugar uptake in Fig 1, as it may be attributed to cell death caused by cherry stems. Providing this information would strengthen the validity of the study.
3. Certain expressions throughout the manuscript appear casual and may impact the seriousness of the scientific content. For instance, in Method 2.3 and Fig 4 legends, the labeling of the drug volume is unconventional. It is suggested to use terms like "concentration" or other scientifically appropriate descriptors for clarity and precision.
I believe addressing these concerns will significantly enhance the manuscript's overall quality and contribute to its scientific rigor.
Author Response
Thank you very much for your evaluation of our manuscript.
We will next answer your comments.
Best regards.
Fátima Martel, corresponding author
- The use of q-PCR to assess glucose transporter mRNA expression is noted. However, to comprehensively explain the glucose uptake phenotype, it is recommended to include analysis at the protein level or evaluate transporter translocation. This additional information would enhance the study's depth and clarity.
Answer: Yes, we agree that analysis at the protein levels would enhance this study. But, in order to perform that, we would need to buy the 3 antibodies and optimize the western and perform the 3 western blots…that would take some time (about 2-3 months). Moreover, we would have to buy another sample of cherry stem, because our infusion is “old”, and we cannot guarantee that the new infusion would be completely identical to the one we have used. I do not know what do you think about this, if it is really crucial to perform these experiments…and also if it is not too late to submit a new version by march 2024.
- Fig 4 lacks supportive data, presenting a significant issue in the manuscript. It is crucial to address this concern by including relevant information. Specifically, the absence of ROS level data after different treatments should be rectified. Without this data, it is challenging to draw conclusions about the observed lower sugar uptake in Fig 1, as it may be attributed to cell death caused by cherry stems. Providing this information would strengthen the validity of the study.
Answer: thank you for this important comment. Indeed, we have tested the effect of the cherry stem infusion (25%(v/v)) on the viability of these cells, because we agree that this is an important point to check. This was published in our previous publication [4], and this was stated in the Introduction (lines 72-74). Anyway, in order to make this point clearer, we added a sentence to the description of the results of Fig. 1 (lines 250-252). Please also noted that uptake results were normalized to the protein content of the wells (lines 158-159); so, differences due to the amount of cells were nullified.
Moreover, we do not think that evaluating ROS levels would strengthen the manuscript, because ROS levels, by themselves, do not give information on the cellular consequences of ROS (because antioxidant system may also be changed by the treatment). This means that we can have an increase in ROS levels without an increase in oxidative stress levels. So, we think that measuring an endpoint of cellular oxidative stress levels, such as MDA levels (TBARS assay), gives a more accurate information on the true oxidative stress levels that the cells are exposed to. In this context, we have previously evaluated the effect of this infusion on MDA levels in Caco-2 cells [4]. This was mentioned in the introduction (lines 66-72). Anyway, in order to make this point clearer, some text was added to the Results section (lines 337-339). Also, “ROS” was replaced by “oxidative stress” in line 340.
- Certain expressions throughout the manuscript appear casual and may impact the seriousness of the scientific content. For instance, in Method 2.3 and Fig 4 legends, the labeling of the drug volume is unconventional. It is suggested to use terms like "concentration" or other scientifically appropriate descriptors for clarity and precision.
Answer: we apologize for this. As a matter of fact there was a mistake in the text, “µl” should have been written “µM”. This has been corrected.
Round 2
Reviewer 2 Report
Comments and Suggestions for Authors
Dear authors,
I appreciate your prompt response to my inquiries. At this point, I have no further comments to add. I recommend that the Editor accept the paper in its current format. However, I still suggest incorporating additional metrics beyond mRNA levels to elucidate your sugar uptake data in your future study.
Thank you!